# Silicon Nano-Fertilizer-Enhanced Soybean Resilience and Yield Under Drought Stress

**DOI:** 10.3390/plants14050751

**Published:** 2025-03-01

**Authors:** Jian Wei, Lu Liu, Zihan Wei, Qiushi Qin, Qianyue Bai, Chungang Zhao, Shuheng Zhang, Hongtao Wang

**Affiliations:** 1School of Agriculture, Jilin Agricultural University, Changchun 130118, China; weijian@ccsfu.edu.cn (J.W.); 14704377775@163.com (Q.B.); zhaochungang1@outlook.com (C.Z.); 18686422696@163.com (S.Z.); 2Changbaishan Key Laboratory of Biological Germplasm Resources Evaluation and Application, Tonghua Normal University, Tonghua 134099, China; liulujlnd@163.com; 3School of Life Sciences, Wuhan University, Wuhan 430072, China; weizihan0011@outlook.com; 4Jilin Changfa Modern Agricultural Technology Group Co., Ltd., Changchun 130118, China

**Keywords:** nano-fertilizer, drought stress, soybean yield, nitrogen fixing potential

## Abstract

Drought stress threatens agriculture and food security, significantly impacting soybean yield and physiology. Despite the documented role of nanosilica (n-SiO_2_) in enhancing crop resilience, its full growth-cycle effects on soybeans under drought stress remain elusive. This study aimed to evaluate the efficacy of n-SiO_2_ at a concentration of 100 mg kg^−1^ in a soil medium for enhancing drought tolerance in soybeans through a full life-cycle assessment in a greenhouse setup. To elucidate the mechanisms of n-SiO_2_ action, key physiological, biochemical, and yield parameters were systematically measured. The results demonstrated that n-SiO_2_ significantly increased silicon content in shoots and roots, restored osmotic balance by reducing the Na^+^/K^+^ ratio by 40%, and alleviated proline accumulation by 35% compared to the control, thereby mitigating osmotic stress. Enzyme activities related to nitrogen metabolism, including nitrate reductase (NR) and glutamine synthetase (GS), improved by 25–30% under n-SiO_2_ treatment compared to the control. Additionally, antioxidant activity, including superoxide dismutase (SOD) levels, increased by 15%, while oxidative stress markers such as hydrogen peroxide (H_2_O_2_) and malondialdehyde (MDA) decreased by 20–25% compared to the control. Furthermore, yield components were significantly enhanced, with pod number and grain weight increasing by 15% and 20%, respectively, under n-SiO_2_ treatment compared to untreated plants in drought conditions. These findings suggest that n-SiO_2_ effectively enhances drought resilience in soybeans by reinforcing physiological and metabolic processes critical for growth and yield. This study underscores the potential of n-SiO_2_ as a sustainable amendment to support soybean productivity in drought-prone environments, contributing to more resilient agricultural systems amidst increasing climate variability. Future research should focus on conducting large-scale field trials to evaluate the effectiveness and cost-efficiency of n-SiO_2_ applications under diverse environmental conditions to assess its practical viability in sustainable agriculture.

## 1. Introduction

Drought stress represents a significant threat to agricultural productivity globally, leading to severe economic losses, particularly in crop yields. The impacts of drought on agriculture are profound, as evidenced by the catastrophic losses experienced during the 2012 drought in the United States, which resulted in damages exceeding $33.9 billion, primarily due to agricultural losses affecting more than 62% of the contiguous United States [1,2]. The cumulative effects of drought on agricultural production can lead to diminished gross domestic production, with estimates suggesting that a single drought event can reduce agricultural GDP by approximately 0.8% globally [3]. Drought conditions not only diminish water availability but also critically impair soil moisture reserves essential for various stages of crop growth, thereby exacerbating yield reductions [2,4]. For instance, wheat, maize, rice, and soybeans are expected to face yield losses of 9–12%, 5.6–6.3%, 18.1–19.4%, and 15.1–16.1%, respectively, if no adaptive measures are implemented [5]. Given the emerging threat of drought to agricultural crops, there is an urgent need for innovative, sustainable, and environmentally friendly solutions that can help farmers adapt to changing climatic conditions, particularly in regions vulnerable to drought stress.

Nano-enabled agriculture represents a transformative approach to addressing modern agricultural challenges, particularly drought stress, which poses significant threats to global food security [6]. One of the primary advantages of nano-enabled agriculture is its ability to enhance nutrient use efficiency through the development of nano-fertilizers. These fertilizers are engineered to control the release of nutrients, ensuring that they are available to plants when needed, thus minimizing losses to the environment and improving overall crop yields [7].

Silicon-based nanomaterials have emerged as a promising tool for enhancing agricultural productivity and resilience in various crops [8]. Their positive effects can be attributed to several mechanisms, including improved nutrient uptake, enhanced stress tolerance, and increased resistance to diseases. One of the primary benefits of Si-based NMs is their role in promoting plant growth and yield. For instance, studies have shown that the application of SiO_2_-NMs significantly enhances physiological parameters such as chlorophyll content and overall biomass in crops like cherry radish, leading to a notable increase in yield [9]. In addition to promoting growth, SiO_2_-NMs play a crucial role in enhancing stress tolerance in plants. Studies indicate that the application of Si and salicylic acid can mitigate the adverse effects of high pH stress in tomato seedlings, thereby improving their growth and biomass [10]. Moreover, SiO_2_-NMs have been shown to improve drought tolerance in wheat by enhancing seed germination and seedling vigor under water-deficit conditions [11]. This stress alleviation is further supported by findings that Si enhances antioxidant defence mechanisms in plants, which is vital for coping with oxidative stress induced by environmental challenges [12]. A previous study examined the effects of SiO_2_-NMs treatments on cucumber growth. Results showed significant impacts on soil soluble Si, water characteristics, yield, and water use efficiency. nSi treatments enhanced soil moisture retention and crop performance [13]. Studies also show that SiO_2_-NMs are more effective than bulk Si in enhancing drought tolerance by improving antioxidant activity and osmolyte metabolism in barley under drought stress [14]. These findings suggest that SiO_2_-NMs offer superior support for plant resilience under drought conditions by effectively modulating key physiological and biochemical processes. Although the role of silicon in mitigating stress is well-documented, detailed mechanistic studies, particularly those examining the full life cycle of soybean under drought stress, remain scarce. This gap in knowledge limits the understanding of silicon’s potential in enhancing drought resilience in soybeans. Soybean (*Glycine max*) is a highly nutritious crop, primarily valued as a high protein source, with growth largely dependent on biological nitrogen fixation [15,16]. This study aims to bridge the gap in understanding the full lifecycle effects of SiO_2_-NMs on soybean growth under drought stress conditions. Prior research has demonstrated that SiO_2_-NMs enhance water retention and nutrient uptake in wheat [17] and improve drought tolerance in maize [18]. Based on these findings, we hypothesized that SiO_2_-NMs will (1) improve drought tolerance in soybeans by enhancing physiological and biochemical responses, (2) optimize crop performance under water-stressed conditions, and (3) positively influence soybean yield. The objectives of this study are to (1) evaluate the impact of SiO_2_-NMs on soybean growth under drought stress, (2) assess changes in key physiological parameters, such as the photosynthesis rate, and (3) determine the effects of SiO_2_-NMs on yield components under drought conditions. Additionally, we analyzed factors such as the activity of key enzymes involved in N-assimilation and nitrogen fixation potential, as well as the levels of antioxidative enzymes. By addressing these key aspects, this study provides crucial insights into the practical application of SiO_2_-NMs for improving soybean yield and promoting sustainable agriculture. This research provides valuable insights for developing sustainable agricultural practices, particularly in drought-prone regions, by enhancing soybean resilience and optimizing yield through SiO_2_-NM applications. By addressing these key aspects, this study provides crucial insights into the practical application of SiO_2_-NMs for improving soybean yield and promoting sustainable agriculture. 

## 2. Results and Discussion

### 2.1. Impact of n-SiO_2_ on Silicon Accumulation and Soil Moisture Retention in Soybeans Under Drought Stress

Soybean plants under DS exhibited significant reductions in Si accumulation in both shoots and roots, leading to decreased resilience and growth. Specifically, DS conditions resulted in a 35% reduction in shoot Si content and a 30% decrease in root Si content compared to control plants. These findings are consistent with studies showing that drought impairs nutrient uptake, particularly Si, which is vital for maintaining plant structural integrity and stress tolerance [19]. The application of n-SiO_2_ under DS conditions markedly improved Si levels in shoots and roots by approximately 40% and 45%, respectively. This enhancement suggests that n-SiO_2_ facilitates Si uptake and translocation even under water-limited conditions, supporting the hypothesis that exogenous Si can mitigate the adverse effects of abiotic stressors [20,21]. Soil Si content in the DS group showed a non-significant decrease of 3% compared to control conditions. However, n-SiO_2_ application increased soil Si levels by 30% relative to DS alone, likely due to enhanced retention in the rhizosphere, which promotes continuous uptake by plants. Improved Si availability reinforces cell walls and enhances plant defenses under stress conditions [22,23]. DS also significantly decreased soil moisture content by approximately 35% compared to control plants. In contrast, n-SiO_2_ treatment under DS improved soil moisture levels by about 20%. This effect may result from enhanced root architecture or improved soil structure, reducing water loss and increasing water availability during drought periods [24,25]. Maintaining soil moisture is crucial for sustaining plant growth and productivity under drought stress [26,27]. These findings highlight the potential of n-SiO_2_ as a strategic amendment to bolster soybean growth under drought conditions. By enhancing Si accumulation and soil moisture retention, n-SiO_2_ treatment significantly improves drought resilience in soybean plants (Figure 1). This aligns with the growing body of literature advocating the use of nanomaterials in agriculture to mitigate the impacts of climate change on crop production [28].

### 2.2. Low-Dose n-SiO_2_ Enhances Soybean Agronomic Traits Under Drought Stress

In our full life cycle study, the application of silicon oxide nanoparticles (n-SiO_2_) demonstrated significant potential in enhancing soybean growth and resilience to drought stress. We reported that n-SiO_2_ promoted key agronomic traits under both optimal and drought-stressed conditions (Figure 2). Under normal conditions, n-SiO_2_ application boosted soybean growth across multiple metrics (Appendix A). For example, plant height increased by 17% compared to the untreated control, while shoot fresh biomass and shoot dry biomass reported notable improvements of 27% and 36%, respectively. Additionally, the total leaf area expanded by 30%, and the number of flowers rose by 25%, indicating n-SiO_2_’s broad growth-promoting effects under ideal growth conditions. Meanwhile, the benefits of n-SiO_2_ application were even more pronounced under drought stress, a condition that typically impedes soybean productivity. In drought-stressed plants, n-SiO_2_ treatment (DS + n-SiO_2_) mitigated adverse effects and preserved growth performance. For example, plant height increased by 20% compared to the drought-stressed control (DS), while shoot fresh biomass and shoot dry biomass improved by 17% and 41%, respectively, indicating effective stress alleviation (Figure 2). Moreover, the total leaf area was 20% larger, and the number of flowers increased by 18% compared to DS, demonstrating that n-SiO_2_ supports reproductive and vegetative growth even under water-limited conditions. Similarly, previous studies have reported the beneficial effects of n-SiO_2_ on plant growth, development, and stress tolerance. For instance, Tamoor et al. (2024) reported that silicon application significantly improved growth parameters, photosynthetic efficiency, and antioxidant activity in rice under stress conditions [29]. Silicon and n-SiO_2_ nanoparticles (Si/SiO_2_NPs) enhance stress resistance by reinforcing physical barriers and activating signaling pathways mediated by phytohormones [30]. Similarly, Sajad et al. (2021) found that Si enhanced agronomic traits in soybean under shading stress [31]. However, limited studies have explored n-SiO_2_ effects on soybean under drought conditions. Our findings underscore the potential of n-SiO_2_ to boost soybean productivity under both optimal and drought-stressed environments, making it a promising strategy to mitigate drought-related yield losses and support resilient crop systems.

### 2.3. n-SiO_2_ Enhances Root and Nodule Physiological Health Under Drought Stress

In this study, the application of n-SiO_2_ demonstrated notable improvements in soybean root and nodule traits under both optimal and drought-stressed conditions. Key root physiological indicators, such as root weight, length, surface area, and nodule formation, showed marked improvements, enhancing the plant’s ability to absorb nutrients and fix nitrogen. For example, under optimal conditions, n-SiO_2_ treatment increased root weight by 21% and root length compared to untreated plants. Even under drought, n-SiO_2_ mitigated stress effects, preserving root weight with a 22% increase over drought-stressed controls and supporting root elongation with a 15% increase (Figure 3). These results suggest that n-SiO_2_ enhances root mass and length, potentially helping roots to explore deeper soil layers for water under limited moisture availability. Similarly, root surface area increased by 30% under regular conditions with n-SiO_2_, while drought-treated plants showed a 25% increase, indicating better water and nutrient uptake capacity. Root diameter followed a similar trend, being 20% thicker under normal and 15% thicker under drought conditions in treated plants, further contributing to drought resilience (Figure 3).

Previously, studies have documented that Si improves root architecture, allowing plants to explore deeper soil layers for water and nutrients, which is particularly beneficial under drought stress. Si can enhance drought tolerance by improving water uptake and regulating stomatal behavior, thereby maintaining plant hydration [32,33]. This is corroborated by studies indicating that Si enhances root surface area and diameter, facilitating better water and nutrient uptake, which aligns with our findings of a 30% increase in root surface area and a 20% increase in root diameter under normal conditions [33]. Moreover, the ability of n-SiO_2_ to mitigate drought stress effects is noteworthy. Our results show a 20% increase in root weight and a 15% increase in root elongation compared to drought-stressed controls (Figure 3). The physiological mechanisms through which Si exerts these effects include the accumulation of osmoprotectants and enhanced root hydraulic conductivity, both of which are critical during periods of water scarcity [32].

Crucially, n-SiO_2_ enhanced nodule health, which is vital for nitrogen fixation [34]. Nodule count and weight rose by 35% and 30% under normal conditions, while under drought stress, n-SiO_2_-treated plants exhibited a 20% and 15% increase in these metrics compared to drought-stressed controls (Figure 3). This preservation of nodule function under water stress suggests that n-SiO_2_ bolsters reproductive aspects of root health, maintaining nitrogen-fixing capacity.

The enhancement of nodule health, as evidenced by the increases in nodule count and weight, is particularly significant for nitrogen fixation. Previously, studies have elucidated the importance of nodule development for effective nitrogen fixation, particularly under stress conditions [35]. The preservation of nodule function under drought stress suggests that n-SiO_2_ not only supports root health but also maintains the symbiotic relationship between legumes and nitrogen-fixing bacteria, which is crucial for sustaining plant growth during adverse conditions [36,37].

### 2.4. Enhancement of Photosynthesis Activity in Soybean Under Drought Stress Induced by n-SiO_2_

Drought stress significantly impaired soybean photosynthetic efficiency, reducing chlorophyll content, photosynthetic rate (Pn), intercellular CO_2_ concentration (Ci), transpiration rate (Tr), and stomatal conductance (Gs). Under DS, total chlorophyll content decreased by 25% compared to the Ctrl, aligning with findings that water deficit conditions disrupt chlorophyll synthesis due to oxidative stress and limited nutrient availability [38]. However, n-SiO_2_ treatment mitigated this effect, boosting total chlorophyll by 15% (*p* < 0.05) under DS, suggesting a protective effect on pigment stability (Figure 4A–C). This improvement in chlorophyll content with n-SiO_2_ likely reflects enhanced stability and synthesis via oxidative damage reduction and improved nutrient uptake [38]. DS reduced Pn by 22% compared to Ctrl, while n-SiO_2_ treatment under DS increased Pn by 18% (*p* < 0.05), indicating enhanced photosynthetic capacity with n-SiO_2_ (Pn, Figure 4D). A 20% drop in Ci under DS was countered by a 20% improvement in n-SiO_2_-treated plants (*p* < 0.05), reflecting enhanced CO_2_ assimilation under stress conditions (Figure 4E). Similarly, Tr declined by 30% under DS, while n-SiO_2_ treatment raised Tr by 15% (*p* < 0.05), potentially aiding water regulation and cooling (Figure 4F).

Under DS, Gs dropped by 40%, limiting water and CO_2_ exchange. n-SiO_2_ treatment restored Gs by 12% compared to DS alone (*p* < 0.05), improving photosynthetic gas exchange (Figure 4G). This increase in Gs highlights n-SiO_2_’s role in facilitating gas exchange, which is critical for photosynthetic efficiency under drought conditions [38]. The boost in Ci and Gs with n-SiO_2_ emphasizes its role in maintaining CO_2_ influx and transpiration, both of which are essential for efficient photosynthesis [38]. Moreover, the improvement in Tr under n-SiO_2_ treatment suggests that silicon helps maintain water balance and cooling during stress, supporting nutrient transport and plant thermoregulation [38]. Similarly, Raza et al. also reported that n-SiO_2_ improved leaf gas exchange and chlorophyll a and b concentrations, though decreased oxidative stress in wheat leaves [39]. Overall, n-SiO_2_ application effectively mitigated drought-induced declines in photosynthetic traits, enhancing chlorophyll stability, photosynthetic rate, and gas exchange. These results demonstrate the potential of n-SiO_2_ to bolster soybean resilience to drought, offering a sustainable solution for crop productivity in water-limited environments.

### 2.5. Modulation of Enzyme Activity, Proline Content, and Na^+^/K^+^ Ratio in Soybean Under n-SiO_2_-Induced Drought Stress

Under drought stress, soybean plants exhibit significant physiological and biochemical shifts that lead to oxidative stress and osmotic imbalance. The marked increase in H_2_O_2_ and MDA levels under DS highlights oxidative damage and lipid peroxidation, with a 40% rise in H_2_O_2_ (Figure 5A–C), indicating substantial cellular stress due to water scarcity [40]. Application of n-SiO_2_ mitigates this effect, reducing H_2_O_2_ by 20% compared to DS alone, suggesting a protective role in maintaining cellular structures under drought (Yan et al., 2016). Similarly, the 35% increase in MDA levels under DS signals enhanced lipid peroxidation (Figure 5C), while n-SiO_2_ treatment lowers MDA by 25%, supporting membrane integrity during drought stress [41]. Drought stress also triggers antioxidant responses to counter elevated reactive oxygen species (ROS). Superoxide dismutase (SOD) activity increased by 30% under DS as an adaptive response, and n-SiO_2_ application further elevated SOD by an additional 15% (Figure 5B), demonstrating its role in boosting the plant’s antioxidant defense system [42]. This enhanced SOD activity with n-SiO_2_ helps scavenge ROS, reducing cellular damage and indicating an improved capacity for oxidative stress management.

In terms of ionic balance, drought significantly disrupted ion homeostasis, as evidenced by a 50% increase in the Na^+^/K^+^ ratio, reflecting an ionic imbalance detrimental to plant function [43]. n-SiO_2_ application effectively restored the Na^+^/K^+^ ratio by 40% (Figure 5A–C), suggesting improved ion regulation and osmotic stability. Proline accumulation, which rose by 60% under DS as an osmoprotectant, was reduced by 35% with n-SiO_2_ treatment (Figure 5E), indicating that n-SiO_2_ reduces osmotic stress, thereby lowering the demand for proline synthesis [44]. Overall, n-SiO_2_ application under drought stress plays a multifaceted role, alleviating oxidative damage, bolstering antioxidant defenses, and restoring osmotic balance in soybean plants. These findings emphasize the potential of n-SiO_2_ to enhance drought resilience in crops, laying the groundwork for further exploration into the mechanisms behind these protective effects. Additionally, future research should explore how n-SiO_2_ may regulate signaling pathways related to osmolyte accumulation and antioxidant defense under drought stress.

### 2.6. n-SiO_2_ Enhances Nitrogen Fixation and Assimilation in Soybean Under Drought Stress

Soybean plants exhibit considerable declines in nitrogen metabolism under DS, indicating substantial impairment in processes critical for growth and productivity. The activities of key nitrogen metabolism enzymes, including NR and NiR, decreased significantly under DS conditions, by approximately 40% and 30% in shoot and root tissues, respectively, compared to control plants (Figure 6A,B) The reduction in enzyme activity under DS conditions is consistent with findings from other studies that highlight the detrimental effects of water deficiency on nitrogen metabolism in legumes [27]. Meanwhile, these reductions highlight the negative impact of water deficiency on nitrogen reduction processes essential for plant growth [45]. However, the application of n-SiO_2_ under DS conditions markedly enhanced NR and NiR activities by 25% and 20%, respectively, relative to DS alone (*p* < 0.05), confirming n-SiO_2_’s role in sustaining nitrogen assimilation during drought stress [46]. Previously, studies have also documented that other nano-based materials play a major role in the nitrogen fixation potential in soybean plants [47,48,49].

Similarly, GS, another crucial enzyme, experienced a 35% decline in activity under DS conditions relative to control plants, which limits nitrogen incorporation into amino acids—a key step in protein synthesis (Figure 6C) [50]. Treatment with n-SiO_2_, however, improved GS activity by 28% over DS alone (*p* < 0.05), indicating that n-SiO_2_ aids in maintaining nitrogen incorporation under drought conditions, supporting protein synthesis essential for plant survival [51]. In addition, glutamate synthase (GOGAT) activity also showed a 40% reduction under DS, which impairs nitrogen assimilation further; n-SiO_2_ application, however, led to a 30% recovery in GOGAT activity compared to DS alone (Figure 6D), suggesting its role in reinforcing nitrogen metabolism under water-limited conditions [52]. Drought stress also impacted urease (UE) activity and nitrogen content, with declines of approximately 45% and 30%, respectively (Figure 6E,F) (Zhang et al., 2020). n-SiO_2_ application not only restored UE activity by 20% but also increased nitrogen content by 18% (*p* < 0.05) over DS alone, which likely enhances nitrogen retention and mitigates ammonia toxicity—both crucial for plant health under drought stress [22]. Notably, n-SiO_2_ also improved nitrogen fixation potential by 33% (*p* < 0.05) under DS conditions (Figure 6G), indicating enhanced symbiotic nitrogen fixation and nitrogen availability during drought [53]. These findings demonstrate the significant impact of n-SiO_2_ on nitrogen metabolism in soybean plants under drought stress by enhancing the activity of nitrogen-related enzymes, increasing nitrogen content, and promoting symbiotic nitrogen fixation. These benefits suggest that n-SiO_2_ enhances drought resilience in soybean, thereby supporting sustainable productivity in water-deficient environments.

### 2.7. Nanoscale SiO_2_ Enhances Soybean Yield and Drought Resilience by Improving Physiological and Metabolic Functions

The application of n-SiO_2_ has emerged as an effective strategy for enhancing drought tolerance in soybean, as shown by significant improvements in yield parameters in this study (Figure 7). Plants under DS displayed notable reductions in pod number, grain weight, and soluble sugar content, which are critical indicators of productivity and metabolic health. Specifically, DS led to a 35% decrease in pod number and a 40% reduction in grain weight per plant (Figure 7), corroborating previous findings that drought negatively impacts reproductive traits and yield in soybean [54,55,56]. n-SiO_2_ application effectively mitigated these adverse effects, boosting yield under both optimal and drought conditions. Under normal conditions, n-SiO_2_ treatment increased pod number and grain weight by 25% and 30%, respectively, while under DS, it led to a 15% increase in pod number and a 20% increase in grain weight compared to DS alone (Figure 7). These findings suggest that n-SiO_2_ enhances physiological resilience, likely through improvements in nitrogen metabolism and photosynthetic efficiency [57,58]. Correlation analysis reinforces the positive effects of n-SiO_2_, showing strong associations between yield components and physiological parameters, such as photosynthetic rate and chlorophyll content (Figure 7). Silicon-enhanced nitrogen metabolism, as indicated by increased enzyme activities, suggests improved nutrient uptake and utilization under stress conditions [59,60,61]. Conversely, oxidative stress markers like hydrogen peroxide (H_2_O_2_) and malondialdehyde (MDA) showed negative correlations with yield, underscoring the detrimental impact of oxidative damage on productivity (Figure 7). n-SiO_2_ application appears to alleviate oxidative stress, thereby supporting improved growth and yield under DS [62,63].

The beneficial effects of n-SiO_2_ are likely due to its role in enhancing water stress management and maintaining metabolic functions. n-SiO_2_ may strengthen cell wall integrity, stimulate root growth, and increase nutrient availability, collectively enhancing drought resilience [64]. Additionally, n-SiO_2_ may promote the synthesis of osmoprotectants like proline, which help maintain cellular water balance during drought [65,66]. These findings align with research on nanomaterials improving drought tolerance in crops such as wheat and rice [67]. Overall, n-SiO_2_ significantly enhances soybean yield and resilience under drought by supporting nitrogen metabolism, photosynthesis, and osmotic balance.

## 3. Materials and Methods

### 3.1. Characterization of Nanoscale Silicon Oxide

Silicon oxide nanoparticles (n-SiO_2_) (99.99%, 17 ± 1.8 nm) were purchased in powder form from “Shanghai Pantian Powder Material Co., Ltd. (Shanghai, China)” The morphology and primary size of n-SiO_2_ were characterized using a transmission electron microscope (Hitachi H-7650, Hitachi Corp., Tokyo, Japan). The zeta potential and hydrodynamic sizes of n-SiO_2_ in DI water (100 mg L^−1^) were measured using a Zetasizer (Nano ZS90, Malvern, UK) (Appendix A).

### 3.2. Experimental Design

The n-SiO_2_ exposure experiments on soybeans were conducted in a greenhouse under controlled environmental conditions at Jilin University (Changchun, China). Surface soil (0–20 cm) for the pot experiment was collected from an experimental station at Jilin Agriculture University. The collected soil was air-dried and sieved through a 2 mm mesh. The physicochemical properties of the experimental soil are presented in Appendix A. The prepared soil was thoroughly mixed with n-SiO_2_ to achieve an initial concentration of 100 mg kg^−1^. Each pot was filled with 2.0 kg of prepared soil. Soil without any n-SiO_2_ addition (0 mg kg^−1^) was used as a control (Ctrl). Soybean (*Glycine max*) seeds (Zhonghuang 13) were purchased from “Shouguang Seeds & Seedling Co., Ltd. (Shouguang, China)”, The seeds were sterilized with 5% (*v*/*v*) sodium hypochlorite for 5 min and rinsed with DI water. The sterilized seeds were placed on moist filter paper soaked in DI water in a tray and germinated in an incubator in the dark at 25 °C for five days. Afterward, uniform-sized seedlings were selected, and each seedling was carefully planted into a pot containing soil. Each pot contained one soybean plant. The pots containing seedlings were then placed in a greenhouse with a day/night temperature of approximately 25 ± 2 °C/25 ± 2 °C and a humidity of around 70%. The Crtl and n-SiO_2_-treated groups were watered every two days from the top with 150 mL of DI water to maintain 70% soil moisture content, while drought stress (DS) and DS + n-SiO_2_ groups were watered with 70 mL of DI water to maintain 35% soil moisture content. Throughout the experiment, soil moisture content was monitored using a HydroSense II Hand-held Soil Moisture Sensor (Campbell Scientific, Logan, UT, USA).

Thirty days after sowing, soybean plants were harvested, and chlorophyll content, enzymatic activities, and nitrogen fixation potential were measured. The samples were divided into shoots (including leaves and stems), roots, and nodules. Fresh samples were stored at −80 °C for the determination of biochemical indicators, as noted below. After 60 days, the water was replaced with Hoagland nutrient solution to provide the macro- and micronutrients to the soybean plants. The Hoagland solution (Hopebiol, Qingdao, China) was prepared according to the manufacturer guidelines. Briefly, 1.26 g of this product and 0.945 g of Ca(NO_3_)_2_ were mixed in 1.0 L of DI water and autoclaved for 30 min at 115 °C. Another set of soybean plants was harvested 120 days after sowing (DAS) to measure full life cycle endpoints. At harvest, plants were thoroughly rinsed with DI water and 0.1% HNO_3_ to remove any soil particles adhering to the plant surfaces. Root surface area was determined using a scanner-based imaging system. After harvesting, roots were thoroughly washed to remove any soil debris and then scanned using a flatbed root scanner. The scanned images were processed using WinRHIZO software to quantify the total root surface area. The root surface area was expressed in square centimeters (cm^2^). Plant height and root length were measured, and the dry biomass of shoot and root was determined after oven-drying at 105 °C for 1 h, followed by 80 °C for 24 h.

### 3.3. Photosynthetic Activities and Pigments

Photosynthetic activities were measured 30 days after sowing by assessing the intercellular carbon dioxide concentration (Ci), net photosynthesis (Pn), stomatal conductance (gs), and transpiration rate (E) of the 3rd fully expanded leaves of soybean. The measurement was conducted using an open gas exchange system (LI-6400XT; Li-Cor Environmental, Lincoln, NE, USA) between 09:00 and 11:00 AM. The photosynthetically active radiations were set to 1000 μmol/m^2^ s^−1^, and the CO_2_ molar fraction was maintained at 400 μmol mol^−1^. To ensure measurement accuracy, the instrument was recalibrated to ensure stable and consistent readings. Chlorophyll content (a, b, and total) in soybean leaves was measured by extracting the leaves with 95% ethanol from the 3rd fully expanded leaf. The extracted chlorophyll was analyzed using a plate reader spectrometer (Beckman 640D, Beckman, Brea, CA, USA) at wavelengths of 665 nm, 649 nm, and 470 nm [68].

### 3.4. Enzymatic Activities

Fresh soybean leaves and roots (100 mg) were ground and homogenized with a phosphorus buffer solution (PSB, pH 7.4) at a 1:9 ratio for the measurement of enzymatic activities. The homogenized samples were then centrifuged at 8000–10,000 rpm for ten minutes, and the supernatants were collected for further analysis. Hydrogen peroxide (H_2_O_2_), superoxide dismutase (SOD), malondialdehyde (MDA), proline content, and the Na^+^/K^+^ ratio were measured according to the manufacturer’s instructions (Nanjing Jiancheng Bioengineering Co., Ltd., Nanjing, China) [48].

### 3.5. Nitrogen Assimilation and Fixation Potential

Fresh soybean shoots and roots were collected 30 DAS for the determination of key enzymes involved in N_2_ assimilation activities, including nitrate reductase (NR), nitrite reductase (NiR), glutamine synthetase (GS), and glutamate synthase (GOGAT). These enzymes were measured following the manufacturer’s instructions (Nanjing Jiancheng Bioengineering Co., Ltd., Nanjing, China). Urease (UE) activity in soybean shoots and roots was determined using assay kits according to the manufacturer’s guidelines (Beijing Boxbio Co., Ltd., Beijing, China). Nitrogen content in the soybean shoots and roots was measured using an organic elemental analyzer (Vario EL model, Elementar, Langenselbold, Germany), with samples employed as dry powdered [48].

For nitrogenase activity, an acetylene assay was performed on fresh soybean nodules. Fresh soybean nodules were collected and placed into 50 mL micro-reaction vials. A syringe was used to extract 10 mL of air from the vials, after which 10 mL of acetylene was added. The vials were incubated for 30 min at 30 °C in the incubator. Subsequently, 500 µL of the resulting gas mixture were extracted and analyzed using gas chromatography (Agilent 7890, Agilent Technologies, Santa Clara, CA, USA) to determine ethylene production [69]. Nitrogenase activity was quantified by calculating the ethylene production rate per gram of fresh nodules weight.

### 3.6. Measurement of Si in Plant and Soil

Soybean shoots, roots, and seeds (collected at 120 DAS) were first dried via lyophilization at −48 °C for 72 h using a freeze-dryer (TF-FD-18S, Shanghai, China). After drying, the plant tissues were finely ground into powder. For silicon analysis, approximately 0.2 g of the powdered samples were digested in 3 mL of HNO_3_ and 0.5 mL of H_2_O_2_ in a 75 mL tube using a microwave digestion system (Ultra WAVE, Milestone, Milan, Italy). For soil mineral content analysis, soil samples from the soybean growing area were subjected to acid-wet digestion using aqua regia (3:1, HCl:HNO_3_) in a microwave digestion system. Following digestion, the samples were diluted to a final volume of 25 mL with DI water and filtered through a 0.25 µm PTFE membrane. The concentrations of Si were determined using ICP-MS (DRCII, PerkinElmer, Norwalk, CA, USA).

### 3.7. Soluble Sugar Content Measurement in Seeds

The total soluble sugar content was determined following a previously established protocol. In brief, 0.5 g of the powdered soybean seed sample was homogenized in 80% ethanol and then centrifuged for 20 min at 2000 rpm. The resulting supernatant was combined with 5% phenol and 98% sulfuric acid, followed by incubation in a 30 °C water bath for 20 min. Absorbance was measured at 490 nm using a UV–visible spectrophotometer. Sugar concentration was measured by comparison with a standard curve prepared using a glucose solution of known concentration [70].

### 3.8. Statistical Analysis

A completely randomized design (CRD) was used in the current experiment, with three replicates for each treatment. Data are presented as the mean ± standard deviation (SD). Statistical analyses were performed using Statistic 8.1 software, with significance evaluated through one-way analysis of variance (ANOVA). Treatment means were compared using the least significant difference (LSD) test, with a significance threshold set at *p* < 0.05. Graphical presentation was performed on GraphPad Prism (version 8.0.2) and OriginPro 2024.

## 4. Conclusions

The results demonstrated that n-SiO_2_ significantly increased silicon content in shoots and roots, as well as soil moisture retention, supporting water availability under drought. Nitrogen metabolism enzyme activities, including NR and GS, improved by 25–30% under n-SiO_2_ treatment, enhancing nitrogen assimilation. Antioxidant activity was boosted, with SOD levels increasing by 15%, while oxidative stress markers such as H_2_O_2_ and MDA decreased by 20–25%, indicating reduced cellular damage. Furthermore, n-SiO_2_ restored osmotic balance by reducing the Na^+^/K^+^ ratio by 40% and alleviating proline accumulation by 35%, thus mitigating osmotic stress. Yield components, including pod number and grain weight, were significantly enhanced, with n-SiO_2_ increasing pod number by 15% and grain weight by 20% under drought conditions compared to untreated plants.

## Figures and Tables

**Figure 1 plants-14-00751-f001:**
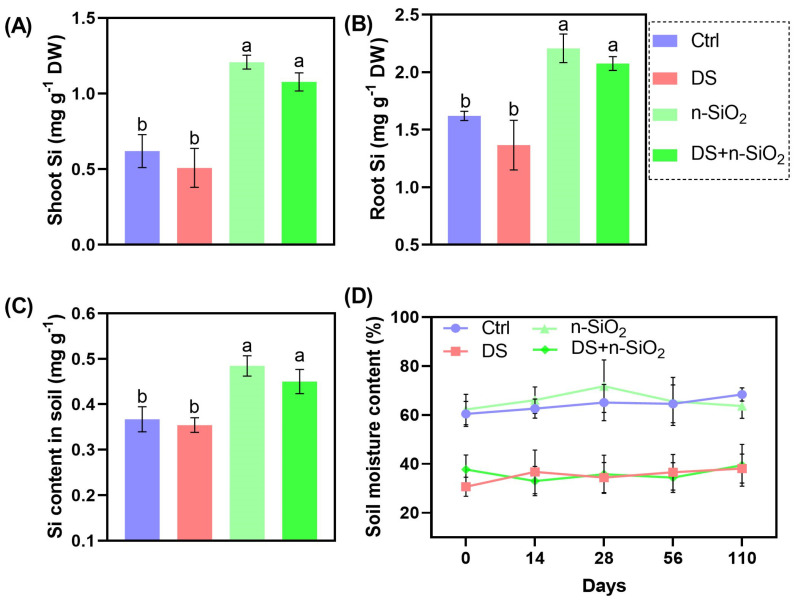
Silicon content in soybean shoots (**A**), roots (**B**), and soil (**C**) under control (Ctrl), drought stress (DS), n-SiO_2_, and combined DS + n-SiO_2_ treatments after the full life cycle. (**D**) Soil moisture content over time across treatments. n-SiO_2_ significantly increased Si content in shoots, roots, and soil under both normal and drought conditions while also enhancing soil moisture retention in n-SiO_2_-treated groups. Different letters indicate significant differences (*p* < 0.05) among treatments.

**Figure 2 plants-14-00751-f002:**
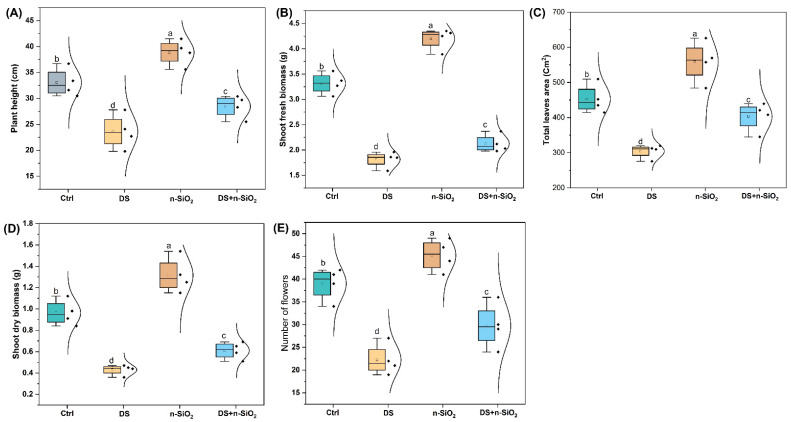
Effects of different treatments on soybean growth parameters at the vegetative stage of the first harvest: control (Ctrl), drought stress (DS), nano-silicon dioxide (n-SiO_2_), and combined drought stress with nano-silicon dioxide (DS + n-SiO_2_). Quantitative analysis of various growth metrics is shown for each treatment group: (**A**) plant height, (**B**) shoot fresh biomass, (**C**) total leaf area, (**D**) shoot dry biomass, and (**E**) number of flowers (flowers data were collected on the 60th day of seedling emergence). Data are presented with different letters to indicate statistically significant differences between treatments (*p* < 0.05).

**Figure 3 plants-14-00751-f003:**
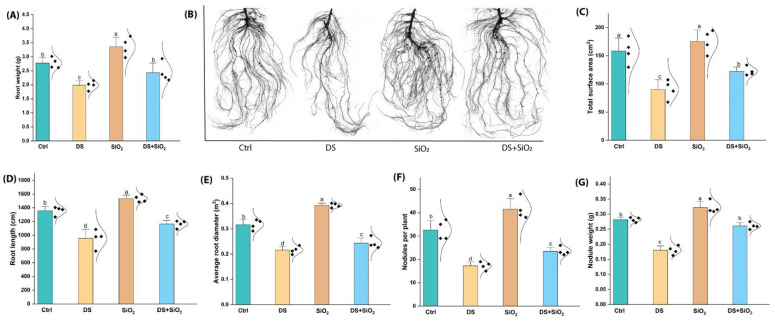
Impact of n-SiO_2_ on soybean root and nodule traits under normal and drought-stressed conditions after the full life cycle. (**A**) Root weight, (**B**) root morphology, (**C**) total root surface area, (**D**) root length, (**E**) average root diameter, (**F**) nodules per plant, and (**G**) nodule weight. Values are expressed as means ± SE, and different letters indicate statistically significant differences among treatments (*p* < 0.05).

**Figure 4 plants-14-00751-f004:**
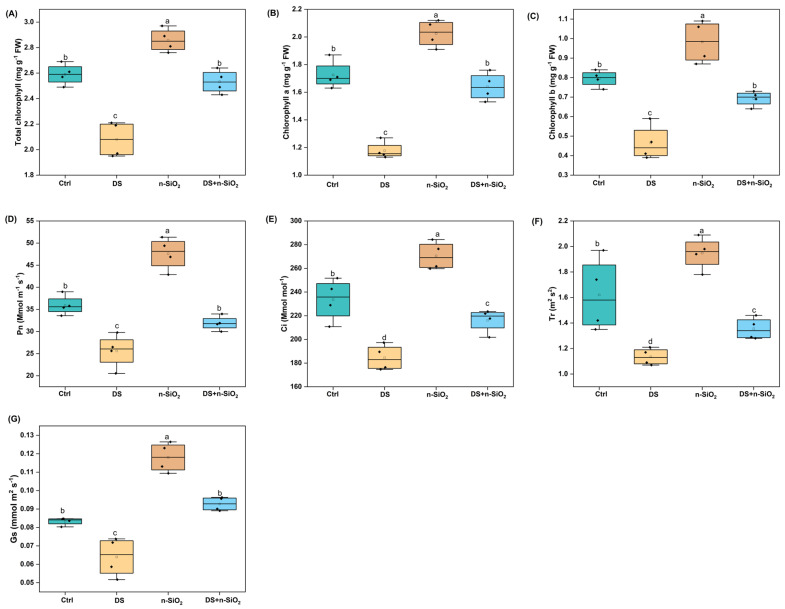
Effects of n-SiO_2_ treatment on soybean root and nodule traits under optimal and drought-stressed (DS) conditions were measured at 28 days. Box plots represent (**A**) total chlorophyll content, (**B**) chlorophyll a content, (**C**) chlorophyll b content, (**D**) net photosynthesis rate (Pn), (**E**) intercellular CO_2_ concentration (Ci), (**F**) transpiration rate (Tr), and (**G**) stomatal conductance (Gs). Different letters above the bars indicate significant differences (*p* < 0.05) among treatments. Ctrl: Control, DS: Drought Stress, n-SiO_2_: Nano-Silicon Dioxide, DS + n-SiO_2_: Drought Stress with Nano-Silicon Dioxide application. The box plots show the median (horizontal line inside the box), the interquartile range (IQR), and whiskers representing the range of data within 1.5 times the IQR. Data points outside this range are considered outliers and are marked with individual dots. Each plot is based on three replicates per treatment, and statistical comparisons were made between the groups.

**Figure 5 plants-14-00751-f005:**
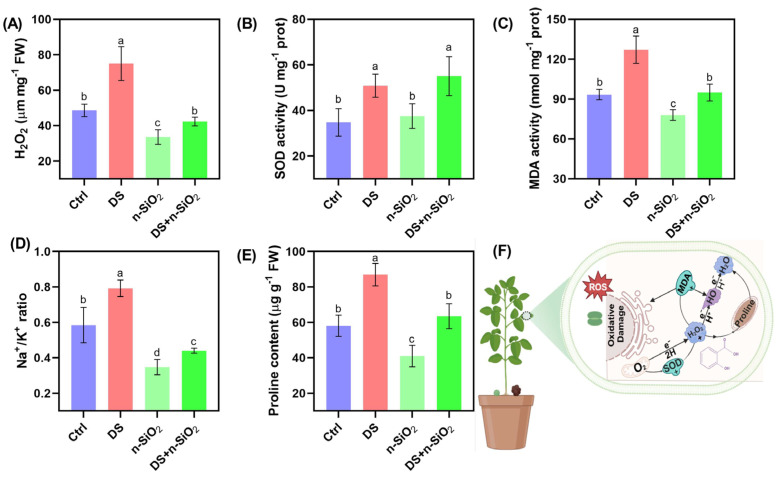
Biochemical responses of soybean under control (Ctrl), drought stress (DS), n-SiO_2_, and combined DS + n-SiO_2_ treatments after 30 days of the experiment. (**A**) Hydrogen peroxide (H_2_O_2_) content, showing a significant increase under DS, which is mitigated by n-SiO_2_ application. (**B**) Superoxide dismutase (SOD), (**C**) malondialdehyde (MDA) levels, (**D**) Na^+^/K^+^ ratio, (**E**) proline content, and (**F**) schematic representation illustrating the role of n-SiO_2_ in alleviating DS. Different letters indicate significant differences (*p* < 0.05) among treatments.

**Figure 6 plants-14-00751-f006:**
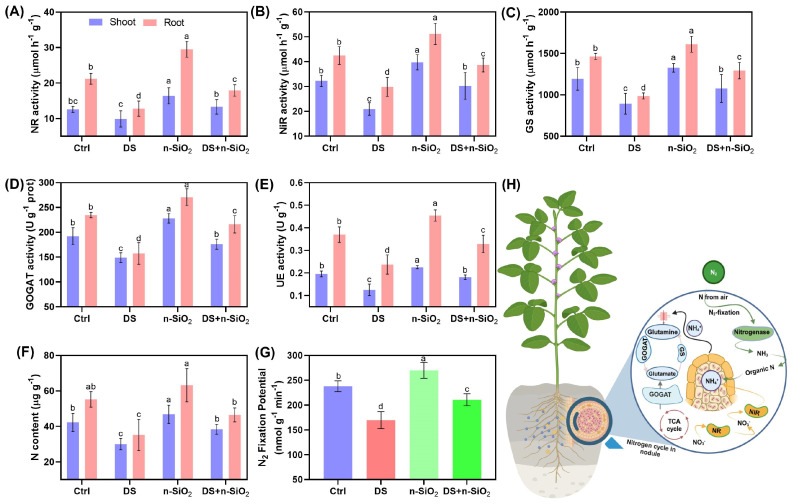
Nitrogen fixation responses in soybean under control (Ctrl), drought stress (DS), n-SiO_2_, and combined DS + n-SiO_2_ treatments after 30 days of the experiment. (**A**) Nitrate reductase (NR) activity in roots and shoots, (**B**) nitrite reductase (NiR) activity, (**C**) glutamine synthetase (GS) activity, (**D**) glutamate synthase (GOGAT) activity, (**E**) urease (UE) activity, and (**F**) nitrogen (N) content in plant tissues, (**G**) N_2_ fixation potential, and (**H**) schematic of nitrogen cycle processes within the soybean root nodule, depicting the contributions of n-SiO_2_ to maintain nitrogen metabolism under drought stress. Different letters indicate significant differences (*p* < 0.05) among treatments.

**Figure 7 plants-14-00751-f007:**
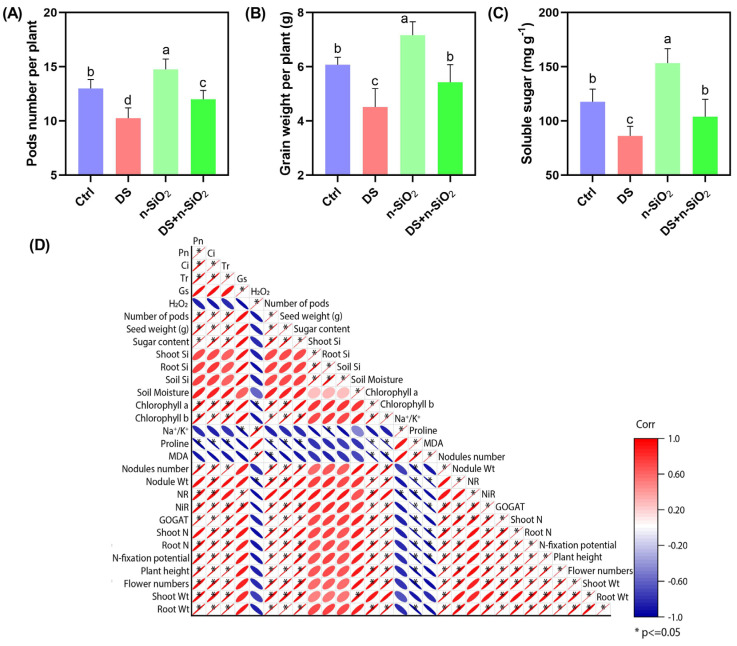
Effects of n-SiO_2_ on soybean yield parameters under control (Ctrl), drought stress (DS), n-SiO_2_, and DS + n-SiO_2_ treatments after a full life cycle. (**A**) Pod number per plant, (**B**) grain weight per plant, and (**C**) soluble sugar content. (**D**) Correlation matrix showing relationships among agronomic, physiological, and biochemical traits, highlighting positive correlations between yield components and photosynthetic and nitrogen metabolism parameters. Different letters indicate significant differences (*p* < 0.05) among treatments.

## Data Availability

All the data is presented in the published paper. Further details can be obtained from the corresponding authors.

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
