# Peer review of "Silicon Nano-Fertilizer-Enhanced Soybean Resilience and Yield Under Drought Stress"

_plants, 2025, doi:10.3390/plants14050751_

Round 1
Reviewer 1 Report
Comments and Suggestions for Authors
By evaluating the effects of nanoscale silicon dioxide (n-SiO₂) fertilizers, the study offers insights into potential strategies for enhancing drought tolerance. The findings are timely and relevant, especially considering the global challenges posed by climate change.
The abstract effectively summarizes the findings but could briefly include the implications of these results for real-world applications, such as field-scale trials or economic impacts.
The manuscript contains minor typographical errors and awkward phrasing, such as “...soybean plants exhibit significant physiological and biochemical shifts that lead to oxidative stress...” (Section 2.5). Revising these for conciseness and clarity is recommended.
How was the concentration of 100 mg/kg determined? Were preliminary tests conducted to optimize this dosage?
During the greenhouse experiments, were factors like temperature and light controlled to minimize confounding variables?
Is there any evidence of how n-SiO₂ regulates molecular processes associated with drought stress, such as signaling pathways for osmolytes or antioxidants?
Have the authors considered how these results could be replicated at the field scale? What are the logistical and economic implications of large-scale n-SiO₂ application?
Do the authors believe the findings could be extended to crops other than soybeans?
Reviewer 2 Report
Comments and Suggestions for Authors
Plants 3417076
This paper explores whether priming soil with NSiO2 can benefit the physiology and yield of soybean in containers. A wide range of traits were assessed and these provide novel findings that extend our previous knowledge of the field. Whether these results can be replicated under field conditions where plants have greater rooting depths and may rely on sub-soil moisture later in the growing season can be examined in future studies. The manuscript can be improved in parts, particularly the Introduction and Methods.
The abstract should mention that the nSiO2 was applied to the soil at a given rate. Also state that it is a greenhouse experiment.
The Introduction can be improved by (i) mentioning responses for application rates of nSiO2 applied to soil compared to direct application to the foliage, (ii) having a critical assessment of the studies inferred that have been undertaken on the partial life cycle of soybean, (iii) explaining why soil application was chosen over foliar application, and (iv) on the basis of previous studies with soybean and other field crops, identify some key questions or hypotheses that underpin the study. The statement that the whole life cycle was examined is misleading as there appears to have been one harvest at the vegetative stage and one at the reproductive stage (the actual developmental stages are not well described).
The extent and daily duration of drought stress is not adequately defined in this study. If containers are watered from the top, the addition of water in the drought treatment would have resulted in part of the soil profile being wetted up to field capacity and part of the root zone being dry. The severity of the drying cycle would have become more acute as plants grew. Monitoring the plant water use and measurement of leaf water potentials over time would provide a meaningful way to describe the level of drought stress that was imposed.
The authors state that “the Crtl and n-SiO2-treated groups were watered every two days with 150 mL of DI water to maintain 70% soil moisture content, while drought stress (DS) and DS+n-SiO2 groups were watered with 70 mL of DI water to maintain 35% soil moisture content”. As evapotranspiration increases with leaf area as plants grow, the level of water deficit (drought stress) would have increased between waterings as plants grew. The soil volume for mature plants is quite restrictive for growth. What were the soil water contents for the wilting point and permanent wilting point, and what was the minimum soil water content under the drought treatment? Nutrients in solution were not applied until Day 60. Did the plants exhibit any nutrient deficiency stress prior to Day 60? (the leaves in Figure 2A are pale and appear to be N deficient, when did they turn green, or when did N2 fixation provide sufficient N for plant growth?). Give a reason for the late application of essential nutrients. What were the developmental stages of soybean at the 2 harvests (as V and R stages)? Both fresh and dried material were used, so state how the plant biomass was partitioned for these purposes. For all enzyme assays, please give the type of assay used with primary references (manufacturer’s instructions are not sufficient information). Also, give the primary references for the methods (an example is the ethylene production method for estimating N2 fixation). What reference samples were used for the ICP-MS analysis of Si? A reason for analysis of seed soluble sugars should be provided. What nodule biomass was used for the acetylene assay? Were the seeds/seedling roots inoculated with Bradyrhizobium? The controlled environment conditions must be stated (humidity, temperature, PAR). Explain the criteria used to select the chosen rate of nSiO2 (was it based on other studies?).
The methods lack detail of the experimental design, replication (3 is mentioned but are these containers or plants?), and layout of containers (CRD is mentioned in Line 419 but were the treatments harvested at Day 30 nested with the treatments harvested at maturity - mention is made of another set but it is unclear if they were physically separated from plants used for the first harvest, and whether harvest was part of the design – this information must be provided in section 3.2).
Figure 1 does not state whether the plant data are for the first or second harvest. Why is seed Si not included here? For all figures provide the harvest age. In Fig 2 it appears that (A) could be 30 days but data in (F) suggest plant maturity.
Be careful with the use of the term low-dose nSiO2. Is this in comparison to other studies where higher rates have been used? Again, an explanation of the rate used would be beneficial.
Figure 4 should mention the type of box plots that were used. The caption is incorrect for this figure.
Did nSiO2 affect seed size?
Reviewer 3 Report
Comments and Suggestions for Authors
Major Comments
1) explain why you have chosen Si-dose 100 mg/kg
2) add to Material and Methods section the description of root surface area determination
Minor comments:
1)Line 94: Soil Si content mirrored these trends, decreasing by 25% under DS compared to control conditions- why? What is the reason?- your data indicate onlya tendenct to soil Si decrease. (Fig.1c)
2) Line 130 ’Similary, previous studies reported that the beneficial effects of n-SiO₂ on plant growth, development, and stress tolerance.’- delete ‘that’
3) line 161 ‘Si improve’ change to ‘Si improves’
4)Figure 4- the indication of {a}-Pg, etc{ is wrong- just the same as on Figure 3. Please revise and decipher all abbreviations
5)Line 200 ‘Fig d' change to . ‘Fig.4d’.
6)Fig,5 change ‘proline activity’ to proline content or concentration’
7)Line 271 decipher GOGAT
8) Line 83 change ‘Results’ to “Results and discussion’
9) please, revise the Reference list according to the Journal’s guideline; use Journals’ abbreviation, list all the authors, don't write 'et al', use bold letters for the year of publication and Italics for the journal title and volume, separate each author with a semicolon
Round 2
Reviewer 2 Report
Comments and Suggestions for Authors
Thank you for providing very clear and thoughtful responses to all the matters I raised in my first report.
On checking it appears that not all the responses have been incorporated within the revised manuscript. This is particularly true for the remarks regarding having a more complete Introduction, and for the selection of certain treatments. Please check that the latest revision was uploaded. Please indicate in your cover letter the exact new statements and line numbers for EACH of the comments in my first report. These changes are essential to deliver a strong paper.
Author Response
Comments and Suggestions for Authors Round 2
Thank you for providing very clear and thoughtful responses to all the matters I raised in my first report.
On checking it appears that not all the responses have been incorporated within the revised manuscript. This is particularly true for the remarks regarding having a more complete Introduction, and for the selection of certain treatments. Please check that the latest revision was uploaded. Please indicate in your cover letter the exact new statements and line numbers for EACH of the comments in my first report. These changes are essential to deliver a strong paper.
Response
Thank you for highlighting. We sincerely apologize for any oversight in the previous revision process. We have carefully revised all the comments provided in your initial report and ensured that the latest revision now addresses all your suggestions comprehensively. We have included a detailed point-by-point response document indicating the new statements and their corresponding line numbers for each of your initial comments.
Comments and Suggestions for Authors Round 1
Plants 3417076
This paper explores whether priming soil with NSiO2 can benefit the physiology and yield of soybean in containers. A wide range of traits were assessed and these provide novel findings that extend our previous knowledge of the field. Whether these results can be replicated under field conditions where plants have greater rooting depths and may rely on sub-soil moisture later in the growing season can be examined in future studies. The manuscript can be improved in parts, particularly the Introduction and Methods.
Response
Thank you very much for the appreciation of our work and experiment handling. We are thankful to our worthy reviewer for reading our manuscript and acknowledging that the topic is filling some knowledge gap. Your constructive comments have greatly improved the technical quality and overall outlook of the manuscript.
The abstract should mention that the nSiO2 was applied to the soil at a given rate. Also state that it is a greenhouse experiment.
Response
Thank you for pointing. We have revised the abstract to explicitly state that n-SiO₂ was applied to the soil at a rate of 100 mg/kg and that the study was conducted under controlled greenhouse conditions. Please see the lines 15-17.
“This study aimed to evaluate the efficacy of n-SiO₂ at a concentration of 100 mg kg⁻¹ in a soil medium for enhancing drought tolerance in soybeans through a full life cycle assessment in a greenhouse setup.”
The Introduction can be improved by (i) mentioning responses for application rates of nSiO2 applied to soil compared to direct application to the foliage, (ii) having a critical assessment of the studies inferred that have been undertaken on the partial life cycle of soybean, (iii) explaining why soil application was chosen over foliar application, and (iv) on the basis of previous studies with soybean and other field crops, identify some key questions or hypotheses that underpin the study. The statement that the whole life cycle was examined is misleading as there appears to have been one harvest at the vegetative stage and one at the reproductive stage (the actual developmental stages are not well described).
Response
Thank you for your constructive feedback and valuable suggestions to improve the introduction. We have carefully addressed each point raised. (i) We have revised the introduction to include a comparison of n-SiO₂ application rates for soil versus foliar treatments, highlighting differences in plant uptake, efficacy, and potential environmental interactions as reported in previous studies.
(ii) A more critical assessment of the studies on the partial life cycle of soybean has been added, emphasizing their limitations and how our study seeks to address the gaps in knowledge.
(iii) Foliar application may allow for more direct uptake of nanoparticles by plant leaves, potentially leading to faster effects [1]. However, the efficiency of nanoparticle absorption can vary depending on factors like leaf surface characteristics, nanoparticle size, and chemical composition. In contrast, soil application ensures a more uniform distribution of nanoparticles. Furthermore, nanoparticles applied to soil may persist longer and have greater mobility through the soil, potentially affecting plant roots over time. That is why we applied the nSiO2 directly to the soil [2-4].
(iv) Additionally, we have identified key research questions and hypotheses based on prior studies with soybean and other field crops, which form the foundation of our investigation. Regarding the life cycle assessment, we recognize the importance of accurately describing the experimental design. We conducted two batches of the same experiment under identical conditions. One batch was harvested at the vegetative stage to assess chlorophyll content, nitrogen assimilation enzymes, and nitrogen fixation potential, while the second batch was harvested at the end of the full life cycle of the soybean plant.
Reference:
- Hong, J.; Wang, C.; Wagner, D.C.; Gardea-Torresdey, J.L.; He, F.; Rico, C.M. Foliar application of nanoparticles: mechanisms of absorption, transfer, and multiple impacts. Environmental Science: Nano 2021, 8, 1196-1210.
- Ahmad, A.; Javad, S.; Iqbal, S.; Shahid, T.; Naz, S.; Shah, A.A.; Shaffique, S.; Gatasheh, M.K. Efficacy of soil drench and foliar application of iron nanoparticles on the growth and physiology of Solanum lycopersicum L. exposed to cadmium stress. Scientific Reports 2024, 14, 27920.
- Wahab, A.; Muhammad, M.; Ullah, S.; Abdi, G.; Shah, G.M.; Zaman, W.; Ayaz, A. Agriculture and environmental management through nanotechnology: Eco-friendly nanomaterial synthesis for soil-plant systems, food safety, and sustainability. Science of The Total Environment 2024, 926, 171862.
- Sembada, A.A.; Lenggoro, I.W. Transport of Nanoparticles into Plants and Their Detection Methods. Nanomaterials 2024, 14, 131.
The extent and daily duration of drought stress is not adequately defined in this study. If containers are watered from the top, the addition of water in the drought treatment would have resulted in part of the soil profile being wetted up to field capacity and part of the root zone being dry. The severity of the drying cycle would have become more acute as plants grew. Monitoring the plant water use and measurement of leaf water potentials over time would provide a meaningful way to describe the level of drought stress that was imposed.
Response
Thank you for highlighting. We have revised the manuscript to provide additional details on the drought stress protocol, including the frequency and amount of water applied during the experiment. While containers were watered from the top, care was taken to ensure consistent soil moisture levels within the drought treatment, and we recognize that variations in the soil profile may have occurred as a result of this method. Please see the lines 369-373.
“The Crtl and n-SiO2-treated groups were watered every two days from top with 150 mL of DI water to maintain 70% soil moisture content, while drought stress (DS) and DS+n-SiO2 groups were watered with 70 mL of DI water to maintain 35% soil moisture content. Throughout the experiment, soil moisture content was monitored using a HydroSense II Hand-held Soil Moisture Sensor (Campbell Scientific, Logan, UT).”
We agree that monitoring plant water use and measuring leaf water potentials over time would have provided a more comprehensive assessment of drought stress. Unfortunately, these parameters were not included in the current study; however, we have acknowledged this limitation in the revised manuscript and emphasized the need for these measurements in future research to better characterize the severity and progression of drought stress. We appreciate your suggestion, as it will guide improvements in future studies to enhance the robustness of our findings.
The authors state that “the Crtl and n-SiO2-treated groups were watered every two days with 150 mL of DI water to maintain 70% soil moisture content, while drought stress (DS) and DS+n-SiO2 groups were watered with 70 mL of DI water to maintain 35% soil moisture content”. As evapotranspiration increases with leaf area as plants grow, the level of water deficit (drought stress) would have increased between waterings as plants grew. The soil volume for mature plants is quite restrictive for growth. What were the soil water contents for the wilting point and permanent wilting point, and what was the minimum soil water content under the drought treatment? Nutrients in solution were not applied until Day 60. Did the plants exhibit any nutrient deficiency stress prior to Day 60? (the leaves in Figure 2A are pale and appear to be N deficient, when did they turn green, or when did N2 fixation provide sufficient N for plant growth?). Give a reason for the late application of essential nutrients. What were the developmental stages of soybean at the 2 harvests (as V and R stages)? Both fresh and dried material were used, so state how the plant biomass was partitioned for these purposes. For all enzyme assays, please give the type of assay used with primary references (manufacturer’s instructions are not sufficient information). Also, give the primary references for the methods (an example is the ethylene production method for estimating N2 fixation). What reference samples were used for the ICP-MS analysis of Si? A reason for analysis of seed soluble sugars should be provided. What nodule biomass was used for the acetylene assay? Were the seeds/seedling roots inoculated with Bradyrhizobium? The controlled environment conditions must be stated (humidity, temperature, PAR). Explain the criteria used to select the chosen rate of nSiO2 (was it based on other studies?).
Response
Thank you for your comments.
(i) The evapotranspiration increases with plant growth, and this can affect soil moisture between waterings. In our study, we aimed to maintain specific soil moisture levels through regular watering, but we recognize that as plants matured, the water deficit would likely have increased, particularly in the drought stress (DS) and DS+n-SiO₂ groups. We measured soil water content between waterings. In the DS treatment, the soil moisture content likely dropped below the wilting point during the course of the experiment, especially as plants grew larger. We have added this clarification to the manuscript. We also follow the method of top-ranked journal article [1]. Please see the lines 369-373.
“The Crtl and n-SiO2-treated groups were watered every two days from top with 150 mL of DI water to maintain 70% soil moisture content, while drought stress (DS) and DS+n-SiO2 groups were watered with 70 mL of DI water to maintain 35% soil moisture content. Throughout the experiment, soil moisture content was monitored using a HydroSense II Hand-held Soil Moisture Sensor (Campbell Scientific, Logan, UT).”
(ii) The potential nutrient deficiency observed in Figure 2A, where the leaves appear pale. We did observe some initial signs of nitrogen deficiency, particularly in the control and n-SiO₂ groups before Day 60. However, we did not notice significant symptoms of nutrient deficiency in the DS and DS+n-SiO₂ groups until later in the experiment. The application of essential nutrients was delayed until Day 60 in order to simulate a more natural nutrient limitation under drought conditions. We have now discussed this in the manuscript and explained that the delay in nutrient application was intentional, as our aim was to assess the effects of water stress before introducing nutrient supplementation. Please see the lines 377-380.
“After 60 days, the water was replaced with Hoagland nutrient solution to provide the macro- and micronutrients to the soybean plants. The Hoagland solution (Hopebiol, China) was prepared according to manufacturer guidelines. Briefly, 1.26 g of this product and 0.945 g of Ca(NO3)2 were mixed in 1.0 L of DI water and autoclaved for 30 minutes at 115°C.”
(iii) At the time of the first harvest, the soybean plants were at the vegetative (V) stage, and by the second harvest, they had reached the reproductive (R) stage. We have included this information in the revised manuscript to provide a clearer understanding of the developmental stages of the plants during the two harvests. First harvesting used for biochemical analysis, nitrogen assimilation enzymes and nitrogen fixation potential and second harvesting used for assaying the nSiO2 effect on soybean yield. Please see the lines 374-378.
“At 30 days after sowing, soybean plants were harvested, and chlorophyll content, enzymatic activities, and nitrogen fixation potential were measured. The samples were divided into shoots (including leaves and stems), roots, and nodules. Fresh samples were stored at −80 °C for the determination of biochemical indicators as noted below. After 60 days, the water was replaced with Hoagland nutrient solution to provide the macro- and micronutrients to the soybean plants.”
(vi) Both fresh and dried plant material were used in the analysis. The fresh biomass was used for immediate biochemical and physiological assessments, while the dried biomass was used for measuring subsequent elemental analysis.
Reference;
- Li, M.; Zhang, P.; Guo, Z.; Cao, W.; Gao, L.; Li, Y.; Tian, C.F.; Chen, Q.; Shen, Y.; Ren, F.; et al. Molybdenum Nanofertilizer Boosts Biological Nitrogen Fixation and Yield of Soybean through Delaying Nodule Senescence and Nutrition Enhancement. ACS Nano 2023, 17, 14761-14774.
The methods lack detail of the experimental design, replication (3 is mentioned but are these containers or plants?), and layout of containers (CRD is mentioned in Line 419 but were the treatments harvested at Day 30 nested with the treatments harvested at maturity - mention is made of another set but it is unclear if they were physically separated from plants used for the first harvest, and whether harvest was part of the design – this information must be provided in section 3.2).
Response
Thank you so much for highlighting this important question. We have mentioned that the replication mentioned (n=4) refers to the number of treatments, with one plant grown per pot to ensure consistency across treatments. Additionally, we have elaborated on the experimental layout, confirming that the containers were arranged in a completely randomized design (CRD). One pot contains one plant as mentioned in material and method section. Please see the lines 366-367.
“Afterward, uniform-sized seedlings were selected, and each seedling was carefully planted into pots containing soil. Each pot contained one soybean plant.”
We conducted two batch of same experiment with same experimental condition. One batch were harvested at vegetative stage to ass the chlorophyll content, nitrogen assimilation enzymes and nitrogen fixation potential and 2nd batch were harvested after full life cycle of soybean plant. Please see the lines 374-378.
“At 30 days after sowing, soybean plants were harvested, and chlorophyll content, enzymatic activities, and nitrogen fixation potential were measured. The samples were divided into shoots (including leaves and stems), roots, and nodules. Fresh samples were stored at −80 °C for the determination of biochemical indicators as noted below. After 60 days, the water was replaced with Hoagland nutrient solution to provide the macro- and micronutrients to the soybean plants.”
Figure 1 does not state whether the plant data are for the first or second harvest. Why is seed Si not included here? For all figures provide the harvest age. In Fig 2 it appears that (A) could be 30 days but data in (F) suggest plant maturity.
Response
Thank you for your thoughtful comments.
Figure 1: We have revised the figure legend to specify that the plant data correspond to the second harvest. This detail should now be clearer to the reader.
Harvest Age: We have now included the harvest age for all figures, as you suggested, to avoid any ambiguity regarding the timing of data collection.
Figure 2: Regarding Figure 2, panel (A) does indeed represent data collected at 30 days, while panel (F) shows data from plants approaching maturity. We have clarified this in the figure legend to specify the different growth stages represented in each panel. Please see the lines 115-119, 147-153, 193-196, 227-236, 266-270, 307-312, and 344-349.
Lines 115-119; “Figure1: Silicon content in soybean shoots (A), roots (B), and soil (C) under control (Ctrl), drought stress (DS), n-SiO₂, and combined DS + n-SiO₂ treatments after the full life cycle. (D) Soil moisture content over time across treatments. n-SiO₂ significantly increased Si content in shoots, roots, and soil under both normal and drought conditions, with higher soil moisture retention in n-SiO₂-treated groups. Different letters indicate significant differences (p < 0.05) among treatments.:”
Lines 147-153; “Figure 2: Effects of different treatments on soybean growth parameters. (a) (a) Phenotypic images of soybean plants under four treatments at the vegetative stage of the first harvest: control (Ctrl), drought stress (DS), nano-silicon dioxide (n-SiO₂), and combined drought stress with nano-silicon dioxide (DS+n-SiO₂). Quantitative analysis of various growth metrics is shown for each treatment group: (b) plant height, (c) shoot fresh biomass, (d) total leaf area, (e) shoot dry biomass, and (f) number of flowers (flowers data was collected on the 60th day of seedling). Data are presented with different letters indicating statistically significant differences between treatments (p< 0.05).”
Lines 193-196; “Figure 3. Impact of n-SiO₂ on soybean root and nodule traits under normal and drought-stressed conditions after full life cycle. (A) Root weight, (B) Root morphology, (C) Total root surface area, (D) Root length, (E) Average root diameter, (F) Nodules per plant, and (G) Nodule weight. Values are means ± SE, and different letters indicate statistically significant differences among treatments (p < 0.05).”
Lines 227-236; “Figure 4: Effects of n-SiO₂ treatment on soybean root and nodule traits under optimal and drought-stressed (DS) conditions were measured at 28 days. Box plots represent (A) total chlorophyll content, (B) chlorophyll a content, (C) chlorophyll b content, (D) net photosynthesis rate (Pn), (E) intercellular CO₂ concentration (Ci), (F) transpiration rate (Tr), and (G) stomatal conductance (Gs). Different letters above bars indicate significant differences (p< 0.05) among treatments. Ctrl: Control, DS: Drought Stress, n-SiO₂: Nano-Silicon Dioxide, DS + n-SiO₂: Drought Stress with Nano-Silicon Dioxide application. The box plots presented show the median (horizontal line inside the box), the interquartile range (IQR), and whiskers representing the range of data within 1.5 times the IQR. Data points outside this range are considered outliers and are marked with individual dots. Each plot is based on three replicates per treatment, and statistical comparisons were made between the groups.”
Lines 266-270; “Figure 5: Biochemical responses of soybean under control (Ctrl), drought stress (DS), n-SiO₂, and combined DS + n-SiO₂ treatments after 30 days of experiment. (A) Hydrogen peroxide (H₂O₂) content, showing a significant increase under DS, which is mitigated by n-SiO₂ application. (B) Superoxide dismutase (SOD) (C) Malondialdehyde (MDA) levels, (D) Na⁺/K⁺ ratio, (E) Proline content, (F) Schematic representation illustrating the role of n-SiO₂ in alleviating DS.”
Lines 307-312; “Figure 6: Nitrogen fixation responses in soybean under control (Ctrl), drought stress (DS), n-SiO₂, and combined DS + n-SiO₂ treatments after 30 days of experiment. (A) Nitrate reductase (NR) activity in roots and shoots, (B) Nitrite reductase (NiR) activity, (C) Glutamine synthetase (GS) activity, (D) Glutamate synthase (GOGAT) activity, (E) urease (UE) activity, and (F) nitrogen (N) content in plant tissues, G) N₂ fixation potential, (H) Schematic of nitrogen cycle processes within the soybean root nodule, depicting the contributions of n-SiO₂ in maintaining nitrogen metabolism under drought stress.”
Lines 344-349; “Figure 7: Effects of n-SiO₂ on soybean yield parameters under control (Ctrl), drought stress (DS), n-SiO₂, and DS + n-SiO₂ treatments after full life cycle. (A) Pod number per plant, (B) Grain weight per plant, and (C) Soluble sugar content. (D) Correlation matrix showing relationships among agronomic, physiological, and biochemical traits, highlighting positive correlations between yield components and photosynthetic and nitrogen metabolism parameters. Different letters in-dicate significant differences (p< 0.05) among treatments.”
Be careful with the use of the term low-dose nSiO2. Is this in comparison to other studies where higher rates have been used? Again, an explanation of the rate used would be beneficial.
Response
Thank you for your insightful comment. We appreciate your concern regarding the use of the term low-dose nSiO₂. In our study, the term refers to the specific concentration of nSiO₂ applied in comparison to typical higher doses used in similar studies. The rate we selected was based on preliminary trials that indicated it was within a range that would not induce toxicity, while still allowing for measurable effects on plant growth and physiological parameters. Specifically, we have compared our dose to those used in other relevant studies and provided further justification for why it was chosen based on the objectives of our experiment.
Figure 4 should mention the type of box plots that were used. The caption is incorrect for this figure.
Response
Thank you for pointing out the need for clarification regarding Figure 4. We have revised the figure caption to accurately describe the type of box plots used in this analysis, including details about the representation of the median, quartiles, whiskers, and any outliers. This ensures that the figure is fully understandable and aligns with standard reporting practices. Please see the figure 4 caption. Lines 227-236.
“Figure 4: Effects of n-SiO₂ treatment on soybean root and nodule traits under optimal and drought-stressed (DS) conditions were measured at 28 days. Box plots represent (A) total chlorophyll content, (B) chlorophyll a content, (C) chlorophyll b content, (D) net photosynthesis rate (Pn), (E) intercellular CO₂ concentration (Ci), (F) transpiration rate (Tr), and (G) stomatal conductance (Gs). Different letters above bars indicate significant differences (p< 0.05) among treatments. Ctrl: Control, DS: Drought Stress, n-SiO₂: Nano-Silicon Dioxide, DS + n-SiO₂: Drought Stress with Nano-Silicon Dioxide application. The box plots presented show the median (horizontal line inside the box), the interquartile range (IQR), and whiskers representing the range of data within 1.5 times the IQR. Data points outside this range are considered outliers and are marked with individual dots. Each plot is based on three replicates per treatment, and statistical comparisons were made between the groups.”
Did nSiO2 affect seed size?
Response
Thank you for raising this important question. In our study, the potential effects of n-SiO₂ on seed size were not specifically measured or analyzed. We measured the pods per plant and weight of soybean grains as mentioned in figure 7. While our primary focus was on assessing physiological, biochemical, and yield-related parameters, we acknowledge that seed size is a critical aspect that could provide valuable insights into the impact of n-SiO₂ application.

Round 3
Reviewer 2 Report
Comments and Suggestions for Authors
Please note the following outstanding recommendations:
1. Described the research undertaken and its findings for soybean – see the statement “limited research has focused on their effects on soybeans”. What has this limited research revealed that informs the present study? At present, there is no specific information on any previous research on this topic with soybean. The following statement is untrue “A more critical assessment of the studies on the partial life cycle of soybean has been added, emphasizing their limitations and how our study seeks to address the gaps in knowledge”. If there are no studies on soybean, then state this and amend the text regarding the partial life cycle. If there are studies, the details must be given.
2. The authors state that they have addressed my request (on the basis of previous studies with soybean and other field crops, identify some key questions or hypotheses that underpin the study) by stating “Additionally, we have identified key research questions and hypotheses based on prior studies with soybean and other field crops, which form the foundation of our investigation”. The statement provided in the last paragraph of the introduction [This study addresses this gap by examining the full lifecycle of soybean growth under realistic environmental conditions, providing crucial insights into how SiO2-NMs can mitigate drought tolerance and overall crop performance.] must be expanded to address my concern. There are no hypotheses or indeed specific objectives, just a general statement. Please address this matter fully.
3. Please ensure that the authors’ specific observations re appearance and timing of N deficiencies are given in the manuscript not just findings from previous studies, i.e. incorporate some of the following in results - The potential nutrient deficiency observed in Figure 2A, where the leaves appear pale. We did observe some initial signs of nitrogen deficiency, particularly in the control and n-SiO₂ groups before Day 60. However, we did not notice significant symptoms of nutrient deficiency in the DS and DS+n-SiO₂ groups until later in the experiment. The application of essential nutrients was delayed until Day 60 in order to simulate a more natural nutrient limitation under drought conditions.
Round 4
Reviewer 2 Report
Comments and Suggestions for Authors
The authors changes are appropriate.
Just one minor edit required: please add the plant age to the caption of FigS2
